# Dynamic Sub-Swarm Approach of PSO Algorithms for Text Document Clustering

**DOI:** 10.3390/s22249653

**Published:** 2022-12-09

**Authors:** Suganya Selvaraj, Eunmi Choi

**Affiliations:** 1Department of Financial Information Security, Kookmin University, Seoul 02707, Republic of Korea; 2Department of Software, College of Computer Science, Kookmin University, Seoul 02707, Republic of Korea

**Keywords:** swarm intelligence, particle swarm optimization, sub-swarm PSO, text document clustering

## Abstract

Text document clustering is one of the data mining techniques used in many real-world applications such as information retrieval from IoT Sensors data, duplicate content detection, and document organization. Swarm intelligence (SI) algorithms are suitable for solving complex text document clustering problems compared to traditional clustering algorithms. The previous studies show that in SI algorithms, particle swarm optimization (PSO) provides an effective solution to text document clustering problems. This PSO still needs to be improved to avoid the problems such as premature convergence to local optima. In this paper, an approach called dynamic sub-swarm of PSO (subswarm-PSO) is proposed to improve the results of PSO for text document clustering problems and avoid the local optimum by improving the global search capabilities of PSO. The results of this proposed approach were compared with the standard PSO algorithm and K-means algorithm. As for performance assurance, the evaluation metric purity is used with six benchmark data sets. The experimental results of this study show that our proposed subswarm-PSO algorithm performs best with high purity comparing the standard PSO and K-means traditional algorithms and also the execution time of subswarm-PSO comparatively takes a little less than the standard PSO algorithm.

## 1. Introduction

In recent years, there has been a tremendous increase in the volume of text documents on the Internet and modern applications which affect the text analysis process (i.e., text feature selection, text document clustering, text categorization, etc.) [1,2]. Text document clustering is an unsupervised classification of textual documents into clusters based on content similarity [3]. The most popular algorithms for clustering are K-means and its variants, as the K-means algorithm is a simple and most used unsupervised partitioning algorithm [4]. Extracting relevant information from the data is a challenging task that needs fast and high-quality text document clustering algorithms. In this study, swarm intelligence (SI) algorithms have been applied to text document clustering and improved one of the best-performing swarm algorithms to improve the quality of results.

SI algorithms include simple unintelligent agents that follow some simple rules to accomplish very complex tasks. SI algorithms are suitable for resolving complex text document clustering problems compared with traditional clustering algorithms [5]. The PSO provides an effective solution to text document clustering [6]. However, the PSO algorithm usually suffers from falling into a premature convergence to local optimum [7]. This is because PSO initializes the particles in starting of the algorithm and the searching behavior includes that all the particles move towards the best solution and search around the local area. In this case, if the initialization does not explore the proper area including the global solution, there is no option to research the undiscovered area to globally search again in the middle of the algorithm. To improve the global searching capability of the PSO algorithm for text document clustering, in this paper, a dynamic subswarm-PSO algorithm is proposed [8]. This proposed algorithm will reinitialize the number of worst fitness particles in each iteration. Six different data sets, created from BBC sports news [9], 20 newsgroups [10], and scientific papers [11], were used in this study. Purity is used as an evaluation metric for this study. The experimental results show that our proposed subswarm-PSO algorithm performs best comparing the standard PSO and K-means traditional algorithms. Also, the average execution time of our proposed algorithm is a little less than the standard PSO algorithm.

The next section provides an overview of the research work focused on the improvements in the PSO algorithm. Section 3 shows the process of the text document clustering problem. Section 4 outlines the existing standard algorithms for text document clustering. Section 5 describes our proposed subswarm approach in the PSO algorithm. Section 7 shows the experimental conditions. Section 8 shows the experimental results and a detailed discussion of these results. Section 9 summarizes the present research work.

## 2. Related Work

This section shows the previous research works (Table 1) related to the PSO algorithm with different sub-swarm optimization techniques.

## 3. Text Document Clustering

This section describes the main process of text document clustering. This process includes text document collection, pre-processing, document representation, clustering, and cluster validation as shown in Figure 1 [18].

Pre-processing is an important step to enhance the performance of the clustering algorithm. As shown in the Figure 1, after collecting the required raw text documents, text pre-processing is used to clean these text documents by applying natural language processing techniques such as tokenization, stop word removal, stemming, and term weighting to delete the unwanted data and manipulate the data. Then, pre-processing turns these clean documents into a t×d term-document matrix. Here, *t* represents the number of unique terms in the document collection, and *d* represents the number of documents. The text document clustering algorithms (PSO and subswarm PSO) directly use this matrix to convert the document data sets into meaningful sub-collections.

As shown in Figure 1, to validate the quality of results from text document clustering algorithms, a few cluster evaluation metrics can be used. The next subsection explains the evaluation metrics in detail.

### 3.1. Evaluation Metrics

This subsection shows the evaluation metrics of text document clustering. Clusters can be evaluated using the metrics such as purity, homogeneity, accuracy, completeness, entropy, F-measure, V-measure, and adjusted rand index (ARI). In this study, we use purity to evaluate the quality of text document clustering algorithms.

#### Purity

Purity measures whether the clusters contain documents from a single category. The purity value ranges from 0 and 1. Purity is computed by dividing the number of properly assigned documents by *N*. The ideal clusters will get a purity value is 1. Equation (Equation 1) is used to compute the purity [19,20]:(1)$$Purity\left(W,C\right)=\frac{1}{N}\sum_{k}max_{j}w_{k}\cap c_{j}$$
where $C=\{c_{1},c_{2},\cdots ,c_{j}\}$ is the set of classes and $W=\{w_{1},w_{2},\cdots ,w_{k}\}$ is the set of clusters.

## 4. Algorithms for Text Document Clustering

This section shows the algorithms such as traditional K-means and standard PSO for the text document clustering problem. The standard PSO is a better performing algorithm than a few popular other standard SI algorithms such as bat and grey wolf optimization for text document clustering [18]. Therefore, in this study, we compare the performance of our proposed subswarm-PSO algorithms with these traditional K-means and standard PSO algorithms.

### 4.1. K-Means Algorithm for Text Document Clustering

The K-means is a popular traditional clustering algorithm that starts with the initialization of a number of clusters k and the random initialization of k cluster centers. After initialization, in the loop, the algorithm assigns each document to its closest centroid based on its similarity and updates the centroid of each cluster. This loop will execute again and again until meets the termination condition as shown in the Algorithm 1 [21].
**Algorithm 1:** K-means text document clustering algorithm
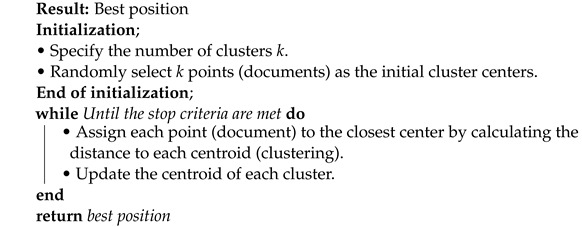


### 4.2. PSO Algorithm for Text Document Clustering

The PSO algorithm is a population-based optimization algorithm developed based on the collective behavior of a flock of birds or fish schools, and aims to locate all the particles in the optimal position [21]. In this study, each particle in the PSO algorithm includes k cluster centroids of text document clusters. The initialization phase is used to initialize the n number of particles $P=\{p_{1},p_{2},\cdots ,p_{n}\}$ in the PSO algorithm and a few PSO parameters. The position of each particle $p_{i}$ will be initialized with k number of cluster centers randomly. After the initialization, k clusters are created by assigning each document to the closest center by calculating the distance to each centroid. Then, the fitness function purity (Equation (Equation 1)) evaluates the clusters, and the global best (gbest) and individual best (pbest) solutions are updated. Then, the new velocity (Equation (Equation 2)) and position (Equation (3)) are calculated for each particle. Until reaching the termination condition iterations will continue as shown in Algorithm 2. As a result, the algorithm finally returns the global best solution [22].
(2)$$v_{i}^{t+1}=\omega *v_{i}^{t}+c_{1}r_{1}^{t}\left(pbest_{i}^{t}-x_{i}^{t}\right)+c_{2}r_{2}^{t}\left(gbest^{t}-x_{i}^{t}\right)$$
(3)$$x_{i}^{t+1}=x_{i}^{t}+v_{i}^{t+1}$$

In Equations (2) and (3), $v_{i}^{t}$ and $x_{i}^{t}$ are the velocity and position of the particle *i* at iteration *t*. $pbest_{i}^{t}$ is the personal or individual best position of *i*. $gbest^{t}$ is the global best position of all particles. ω is the inertia weight to balance the global search and local search. $c_{1}$ and $c_{2}$ are positive constant values used to control the speed of the particle. $r_{1}$ and $r_{2}$ are random parameters within [0,1] [19,21].
**Algorithm 2:** PSO text document clustering algorithm
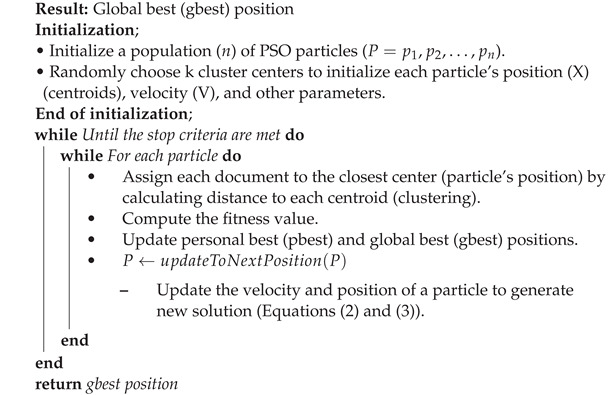


## 5. Proposed Approach

This section describes the proposed subswarm-PSO approach. The main focus of this study is to optimize the result of the PSO algorithm for the text document clustering (Algorithm 2) by applying dynamic sub-swarm techniques to the PSO algorithm as shown in Figure 2.

The initialization section of the proposed subswarm-PSO algorithm is the same as the PSO algorithm and its starts with the initialization of populations (*n*) and other parameters of PSO. Each particle of the PSO algorithm is randomly initialized with the k cluster centers. Then the algorithm computes text document clustering around the cluster centers and executes the fitness function.

The proposed approach is implemented in the PSO algorithm while moving the position of the particles to the next position in the PSO algorithm. Here, based on the fitness solution, we dynamically divide the entire swarm population into two. That is the top n/2 best fitness particles are grouped into one sub-swarm ($SP_{2}$) and the other n/2 particles with the remaining least fitness solutions are grouped into another sub-swarm ($SP_{1}$). As shown in Figure 2, the particles of sub-swarm $SP_{2}$ will move to the next position and update the pbest and gbest solutions among them. The particle solutions from $SP_{1}$ are abandoned and re-initialized randomly. Then, the sub-swarms $SP_{1}$ and $SP_{2}$ will be combined and moved to the next iteration until meets the termination condition.

In this proposed approach, to divide the two sub-swarms, the optimum number n/2 is chosen from the experiment with a different number of abandoned particles with the least solutions such as 1, n/3, n/2, n−(n/3), and n−1 as shown in the experiment and results section.

## 6. Proposed Dynamic Sub-Swarm Pso Algorithm for Text Document Clustering

As shown in the Algorithm 3, subswarm-PSO algorithm initialize n number of particles $P=\left\{p_{1},p_{2},p_{3},\cdots ,p_{n}\right\}$ and other PSO parameters. Each particle in the PSO randomly initializes the k number of centroid points of the clusters. After initialization, the clusters are created by assigning each document to the closest centroid by calculating the distance to each centroid. Next, fitness values will be calculated for each particle. The pbest and gbest solutions will be updated as shown in the algorithm.
(4)SP←sort(P)
(5)$$SP_{1}=\left\{K_{1}|K_{1}\epsilon SP,K_{1}=\left\{s_{1},s_{2},s_{3},\cdot \cdot \cdot ,s_{n/2}\right\}\right\}$$
(6)$$SP_{2}=\left\{K_{2}|K_{2}\epsilon SP,K_{2}=\left\{s_{(n/2)+1},s_{(n/2)+2},s_{(n/2)+3},\cdot \cdot \cdot ,s_{n}\right\}\right\}$$
(7)$$SP_{1}^{{}^{\prime }}=\left\{SP^{{}^{\prime }}\left[j\right]=randomInitialization\left(\right),forj=1,2,\cdots ,n/2\right\}$$
(8)$$SP_{2}^{{}^{\prime }}=\left\{SP^{{}^{\prime }}\left[j\right]=updateToNextPosition\left(SP\left[j\right]\right),forj=\left(n/2\right)+1,\left(n/2\right)+2,\cdots ,n\right\}$$
(9)$$P^{{}^{\prime }}=SP_{1}^{{}^{\prime }}\cup SP_{2}^{{}^{\prime }}$$
where particles of PSO $P=\left\{p_{1},p_{2},p_{3},\cdots ,p_{n}\right\}$, sorted particles $SP=\left\{s_{1},s_{2},s_{3},\cdots ,s_{n}\right\}$, and i=1,2,3,…,n.

Then, before updating the particles and moving to the next iteration in Algorithm 2, our proposed approaches are applied (from Equation (Equation 4) to (9)) in the PSO algorithm as shown in Algorithm 3. Here, Equation (Equation 4) and Algorithm 3 show that all particles P of PSO are sorted by fitness values using quicksort in ascending order and stored in SP. The total sorted swarm population is divided based on the fitness solution into two sub-swarms $SP_{1}$ and $SP_{2}$ as shown in Equations (5) and (6) respectively.

The sub-swarm $SP_{1}$ includes particles with the least n/2 fitness solutions and $SP_{2}$ includes particles with top n/2 fitness solutions. Here, we abandon the solutions of the sub-swarm particles $SP_{1}$ and randomly reinitialized these particles as shown in Equation (7). As shown in Equation (Equation 8), the sub-swarm $SP_{2}^{{}^{\prime }}$ includes the particles of $SP_{2}$ with updated new velocity (Equation (Equation 2)) and position (Equation (Equation 3)). The sub-swarms $SP_{1}^{{}^{\prime }}$ and $SP_{2}^{{}^{\prime }}$ will be combined as $P^{{}^{\prime }}$ as shown in Equation (Equation 9) and assigned to particles *P* for the next iteration as shown in the Algorithm 3. The final result of the algorithm includes the global best solution.
**Algorithm 3:** Dynamic Sub-Swarm Approach of PSO Algorithms for Text Document Clustering
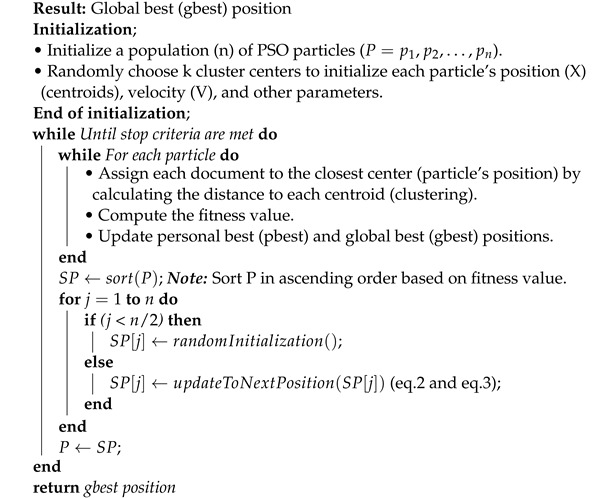


## 7. Experiment

This section shows the details of benchmark data sets and experimental conditions that were used for this experiment.

### 7.1. Benchmark Data Sets

In this study, six different benchmark machine learning data sets were used that are constructed from BBC sports news [9], 20 newsgroups [10], and scientific papers [11]. These data sets are most commonly used for text document clustering and are publicly available for machine learning research [23,24]. Considering the computation time and our resources, we have used only six different data sets for this study.

As shown in Table 2, data set 1 includes 1427 documents and 23,057 terms and these documents belong to 2 clusters (alt.atheism and talk.religion.misc) from 20 newsgroups. Data set 2 consists of 737 documents and 4613 terms and belongs to 5 clusters (athletics, cricket, football, rugby, and tennis) from BBC sports news. Data set 3 includes 40 documents and 2596 terms and belongs to 5 clusters (athletics, cricket, football, rugby, and tennis) from BBC sports news. Data set 4 includes 200 documents and 8716 terms and these documents belong to 4 clusters (rec.motorcycles, rec.sport.hockey, sci.electronics, and talk.religion.misc) from 20 newsgroups. Data set 5 includes 100 documents and 5549 terms and these documents belong to 3 clusters (rec.motorcycles, rec.sport.hockey, and talk.religion.misc) from 20 newsgroups. Data set 6 includes scientific papers in the field of computer science. The data set contains 675 scientific papers and 27,416 terms under 4 classes including case-based reasoning (CBR), inductive logic programming (ILP), information retrieval (IR), and sonification (SON). Here, each paper is including the title, authors, abstract, and references.

### 7.2. Experimental Conditions

In this study, the population size of each SI algorithm is assigned as 10. Here, for each algorithm, we performed experiments for the iterations such as 10, 20, 30, …, 100. Five simulations are used for each iteration number to handle the stochastic results of algorithms. The performance of this algorithm is evaluated using the evaluation metric purity and these results were compared with the algorithms K-means and PSO.

In this experiment, our proposed approach is applied with a different number of abandoned particles with the least solutions such as 1, n/3, n/2, n−(n/3), and n−1 to find the optimal number of abandoned particles and with the optimal number, the comparative study is performed. To determine the optimum number of particles to be abandoned, we use experiments with 10 and 20 particles.

Here, to compare the performance of PSO and proposed subswarm-PSO, the same default parameters ω = 0.9, $c_{1}$ = 0.5, and $c_{2}$ = 0.3 of the PSO algorithm were used in both algorithms from the literature [18,25].

## 8. Results

This section shows the experimental results of finding the optimum number of abandoned least solutions of particles and the comparative results of the proposed subswarm-PSO algorithm with the optimum number of abandoned particles with the least solution for the text document clustering problem.

### 8.1. Different Number of Abandoned Particles with Least Solutions for Subswarm-PSO

This section shows the experimental results of a different number of abandoned particles of PSO for the subswarm-PSO approach to find the optimum number of abandoned particles with the least solution. Here, we experiment and compare with 1, n/3, n/2, n−(n/3), and n−1 number of abandoned particles for subswarm-PSO with data set 5 for 10 and 20 particles.

This experiment includes the following.

1 particle with the least solution will be abandoned and reinitialized randomly. The remaining n−1 particles will move to the next position.n/3 particles with the least solution will be abandoned and reinitialized randomly. The remaining n−(n/3) particles will move to the next position.n/2 particles with the least solution will be abandoned and reinitialized randomly. The remaining n/2 particles will move to the next position.n−(n/3) particles with the least solution will be abandoned and reinitialized randomly. The remaining n/3 particles will move to the next position.n−1 particles with the least solution will be abandoned and reinitialized randomly. The remaining 1 particle will move to the next position.

Here, Table 3 and Figure 3 show the comparative experimental results for a number of abandoned particles with the least solutions such as 1, n/3, n/2, n−(n/3), and n−1 are compared with the standard PSO algorithm for 10 particles. This table shows the maximum, mean, and standard deviation of purity values for the iterations (1, 10, 20, 30, …, 100) and data set 5. This table also shows the average and ranks for all numbers of abandoned particles. Here, the results show that n/2 abandoned particles show the highest performance with the purity values 0.806, 0.812, 0.816, 0.836, 0.842, 0.848, and 0.850 for iterations 10, 30, 40, 60, 70, 90, and 100, respectively. The next level performance is shown for n−(n/3) and n/3 with an average of 0.797 and 0.788.

Figure 3 shows that comparing the standard PSO algorithm with other compared methods shows better performance and half-abandoned (n/2) particles show the best performance.

Here, Table 4 shows the comparative experimental results for a number of abandoned particles with the least solutions such as 1, n/3, n/2, n−(n/3), and n−1 are compared with the standard PSO algorithm for 20 particles. This table shows the maximum, mean, and standard deviation of purity for the iterations (1, 10, 20, 30, …, 100) and data set 5. This table also shows the average and ranks for all numbers of abandoned particles. Here, the results show that n/2 abandoned particles show the highest performance with the purity values 0.862, 0.852, 0.866, and 0.906 for iterations 20, 40, 70, and 90 respectively. The next level’s performance is shown for n/3 with an average of 0.832. Figure 4 shows the results of a different number of abandoned particles for 20 particles and this figure shows that comparing the standard PSO algorithm other compared methods show better performance and half-abandoned (n/2) particles show the best performance with the highest average value of 0.843.

The results from Table 3 and Table 4 show that experiments with 10 and 20 particles are having almost similar patterned results and half-abandoned (n/2) particles show the best performance.

### 8.2. Performance Comparison of Proposed Subswarm PSO Algorithm

This section shows the performance comparison results of the proposed subswarm-PSO algorithm with an optimal number of abandoned particles n/2 with the least solution which is determined by the experimental results of the previous subsection.

Table 5 shows the maximum, mean, and standard deviation of purity values for the text document clustering algorithms. These values are the average of all the iteration numbers. The highest purity means values among all the algorithms for each data set are highlighted in bold. Here, the subswarm-PSO algorithm has the highest purity mean values of 0.728, 0.820, 0.650, 0.807, and 0.888 for data set 1, 3, 4, 5, and 6 respectively. The algorithm K-means shows the highest purity mean value of 0.763 for data set 2. The K-means algorithm shows the lowest purity mean values of 0.527, 0.651, 0.450, and 0.595 for data sets 1, 3, 4, and 5 respectively.

Table 6 shows the ranking of the mean of purity values for text document clustering algorithms with all data sets for the data shown in Table 5. This table shows that the total performance ranks of the algorithms K-means, PSO, and subswarm-PSO are 3, 2, and 1 respectively.

The comparative results and average running time of all algorithms for the data sets 1 to 6 are shown in the Figure 5, Figure 6, Figure 7, Figure 8, Figure 9, Figure 10, Figure 11, Figure 12, Figure 13, Figure 14, Figure 15 and Figure 16. K-means takes very little time for execution. So, we ignored K-means and compared the execution time for only PSO and subswarm-PSO.

Figure 5 shows the comparative results of the proposed algorithm subswarm-PSO with all other algorithms for data set 1. Here, the result shows that the proposed algorithm shows the best performance than other algorithms and the algorithm PSO is performing second. K-means is the lowest-performing algorithm. Figure 6 compares the average execution time of PSO and subswarm-PSO for the results of Figure 5 and Table 5. Here, the proposed algorithm subswarm-PSO takes 5.1% less time than PSO for the execution on average.

Figure 7 shows the comparative results and Figure 8 show the average execution time of the proposed subswarm-PSO algorithm with other algorithms for the data set 2. From Figure 7 we can see that comparing other algorithms, K-means is performing well. However, our proposed algorithm subswarm-PSO shows a second better performance than PSO algorithms. On average our proposed algorithm subswarm-PSO takes 0.8% less time for the execution than PSO as shown in Figure 8.

Figure 9 and Figure 10 show the comparative results and average running time respectively for all algorithms for data set 3. Figure 9 and Table 5 show that our proposed subswarm-PSO algorithm shows the highest purity. The PSO algorithm performs second. Here, the K-means algorithm shows the least performance. Figure 10 shows that subswarm-PSO takes 13.7% less execution time than PSO.

Figure 11 and Figure 12 are the comparative results and average running time respectively of all algorithms for data set 4. From Figure 11 and Table 5, we can say that the proposed subswarm-PSO shows the highest performance, and the PSO algorithm performs second. K-means shows the least performance for dataset 4. Figure 12 shows similar results that subswarm-PSO take 2.1% little less execution time than PSO.

Figure 13 and Figure 14 are the comparative results and average running time respectively of all algorithms for data set 5. Figure 13 shows that subswarm-PSO, PSO, and K-means algorithms perform first, second, and third respectively. Here, similarly to other data sets subswarm-PSO takes 0.75% less execution time than PSO.

Figure 15 and Figure 16 are the comparative results and average running time respectively of all algorithms for data set 6. Figure 15 shows that subswarm-PSO, PSO, and K-means algorithms perform first, second, and third respectively. Here, the algorithm subswarm-PSO takes 4.31% less time for execution than PSO as shown in Figure 16.

### 8.3. Discussion

Based on the above results, we can say that the proposed algorithm subswarm-PSO is performing best to find the optimal solution for most of the data sets by comparing standard PSO and K-means algorithms in text document clustering. The average running times of this proposed subswarm-PSO algorithm are 5.1%, 0.8%, 13.7%, 2.1%, 0.75%, and 4.31% lesser than the standard PSO algorithm for data sets 1, 2, 3, 4, 5, and 6 respectively. The K-means is much faster than other algorithms but it shows the very least optimum solutions for all data sets except data set 2.

The standard PSO algorithm has more ability to solve complex optimization problems such as text document clustering but is usually trapped into the local optimum. Our proposed algorithm ignores half of the least solutions and reinitializes the particles which include the least solution in each iteration. This proposed approach increases the global search capability of PSO and enhances the performance of the PSO algorithm. During each iteration, only half of the particles which include the best solutions are moved to the next solution, and others are reinitialized randomly. Random initialization takes less time compared with updating particles to the next position. This might be the reason for the reduced execution time of our proposed algorithm.

In our proposed method, we choose half (n/2) of the particles with the least solution to abandon based on the results from Table 3 and Table 4. Comparing the standard PSO with other different numbers of abandoned solutions, this half (n/2) and nearly half n/3 and n−(n/3)) are showing better performance. While choosing n/2, it may be balanced well to retain the top half of good solutions to move towards the global best solution and use the remaining half of the particles to explore the new solution. Here, if we reduce the number of abandoned particles to n/2, the global search capability may be reduced, and if increase the number of abandoned particles to more than n/2, may be distracted from moving toward the global best solution. This might be the reason for the n/2 abandoned particles performing well.

## 9. Conclusions

The standard PSO algorithm is more effectively solving complex problems like text document clustering compared with other standard SI algorithms and traditional K-means algorithms. Still, PSO needs to be improved to avoid the problems such as premature convergence and local optima. In this study, we propose a subswarm-PSO algorithm to increase the global search capabilities of PSO and avoid the local optimum to improve the results for text document clustering. Our proposed algorithm is to select potential solutions and re-initialize particles dynamically. Out of a number of experimental observations, selecting half of the best solutions in each iteration has provided performance achievement. The results of this proposed approach were compared with the standard PSO algorithm and the K-means algorithm. Here, evaluation metric purity is used with six benchmark data sets. The experimental results show that our proposed algorithm subswarm-PSO performs best and uses less execution time than the standard PSO algorithm. Previous literature shows that the PSO is also performing well in a few other problems such as clustering, feature selection, and scheduling [18]. Our proposed approach also may perform well in those problems as the PSO algorithm. As a future study, we are aiming to apply and check the performance of the proposed subswarm-PSO approach in different domains.

## Figures and Tables

**Figure 1 sensors-22-09653-f001:**
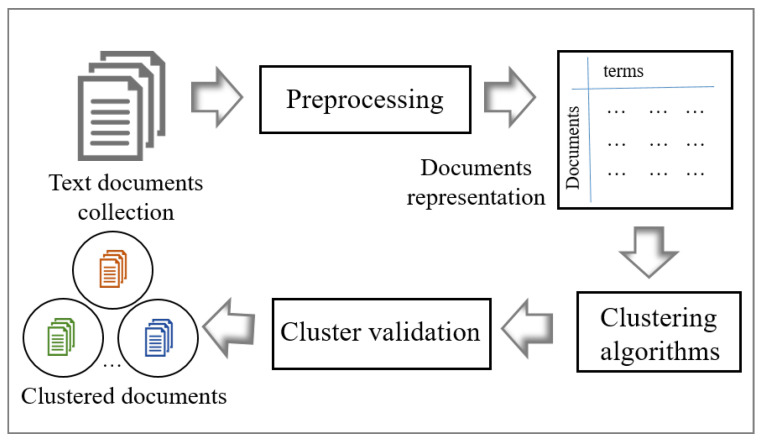
The Process of Text Document Clustering [18].

**Figure 2 sensors-22-09653-f002:**
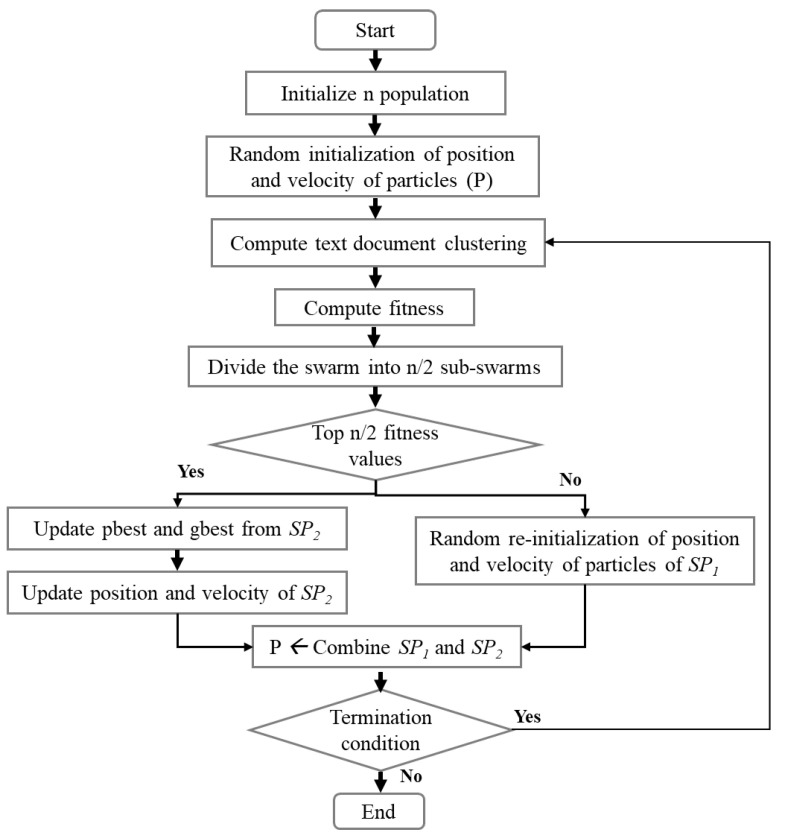
The proposed dynamic sub-swarm for PSO algorithm.

**Figure 3 sensors-22-09653-f003:**
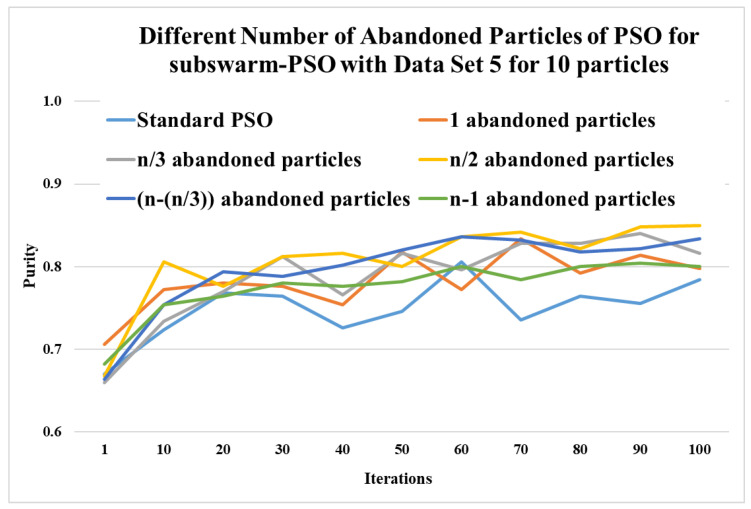
Comparison of different numbers of abandoned particles of subswarm-PSO for data set 5 with 10 particles.

**Figure 4 sensors-22-09653-f004:**
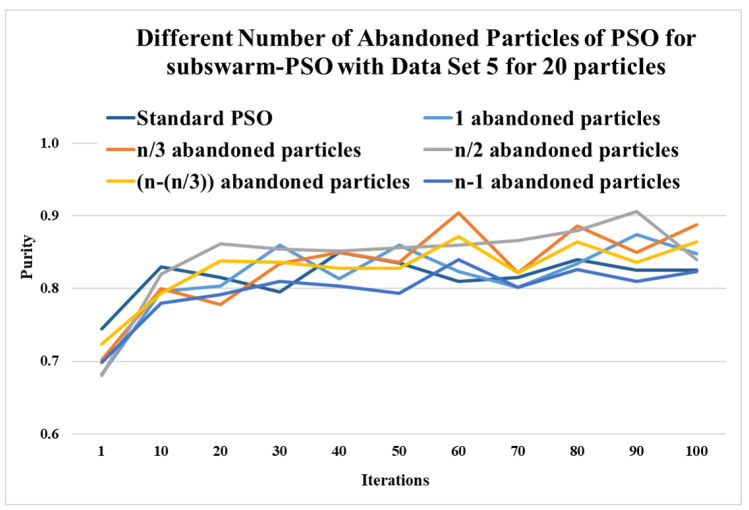
Comparison of the different number of abandoned particles of subswarm-PSO for data set 5 with 20 particles.

**Figure 5 sensors-22-09653-f005:**
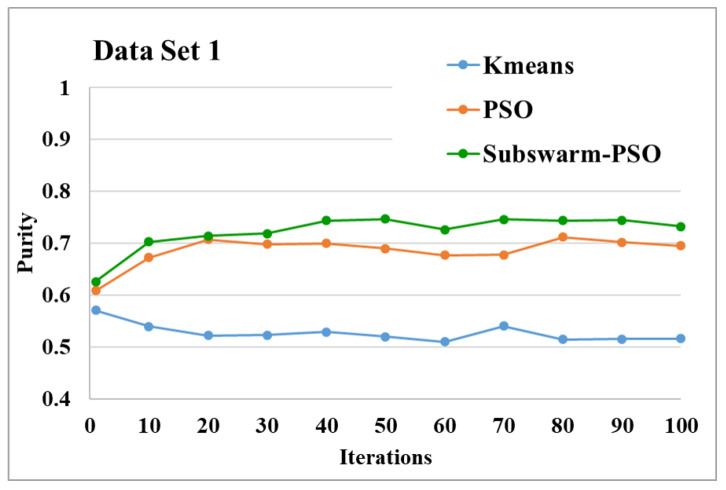
Purity comparison for data set 1.

**Figure 6 sensors-22-09653-f006:**
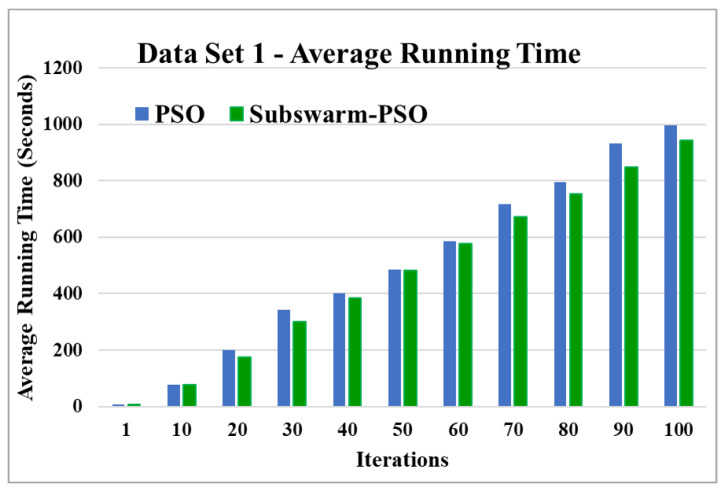
Average running time for data set 1.

**Figure 7 sensors-22-09653-f007:**
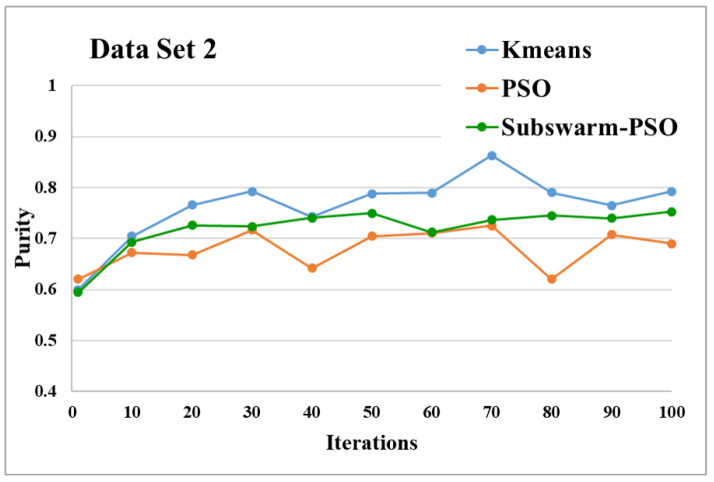
Purity comparison for data set 2.

**Figure 8 sensors-22-09653-f008:**
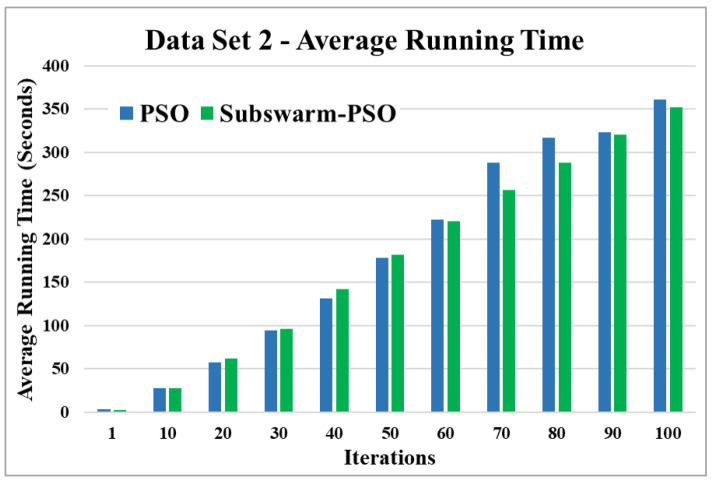
Average running time for data set 2.

**Figure 9 sensors-22-09653-f009:**
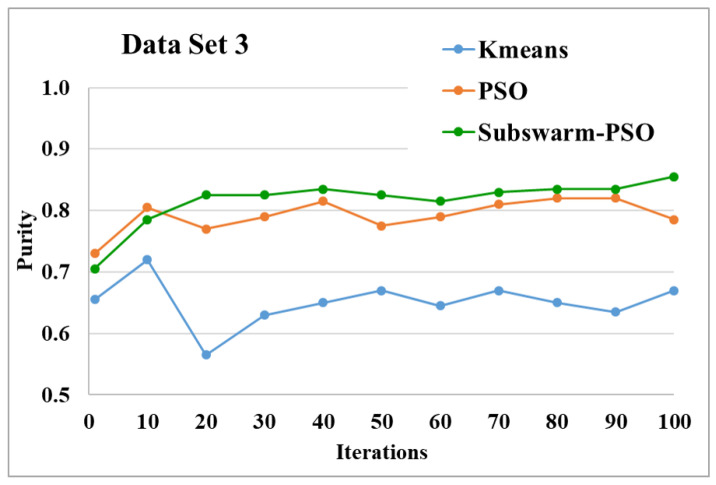
Purity comparison for data set 3.

**Figure 10 sensors-22-09653-f010:**
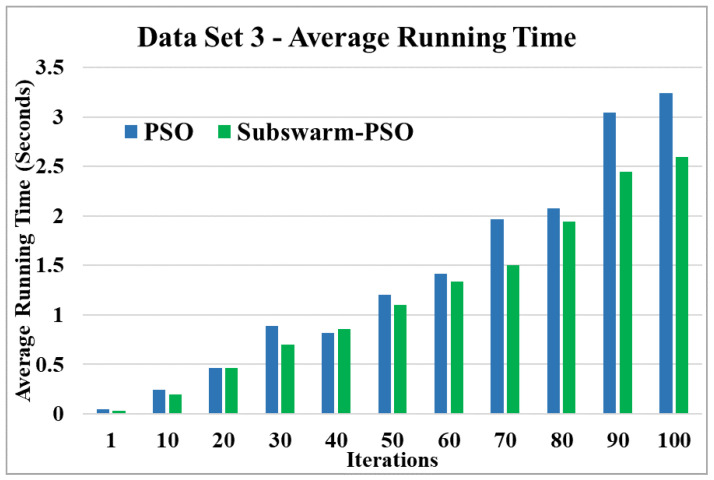
Average running time for data set 3.

**Figure 11 sensors-22-09653-f011:**
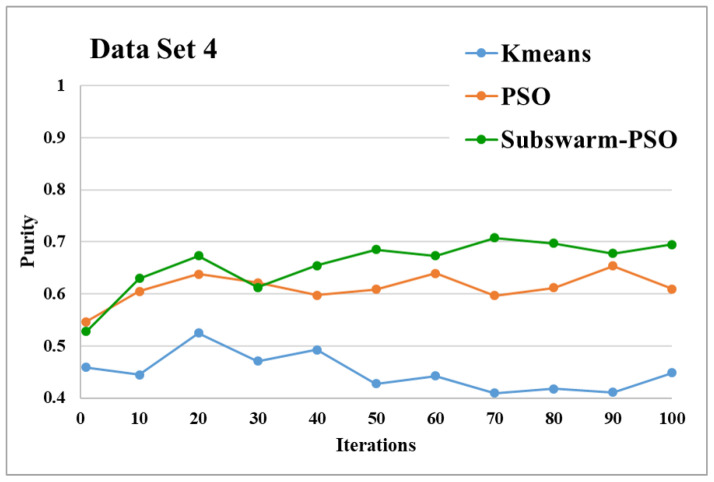
Purity comparison for data set 4.

**Figure 12 sensors-22-09653-f012:**
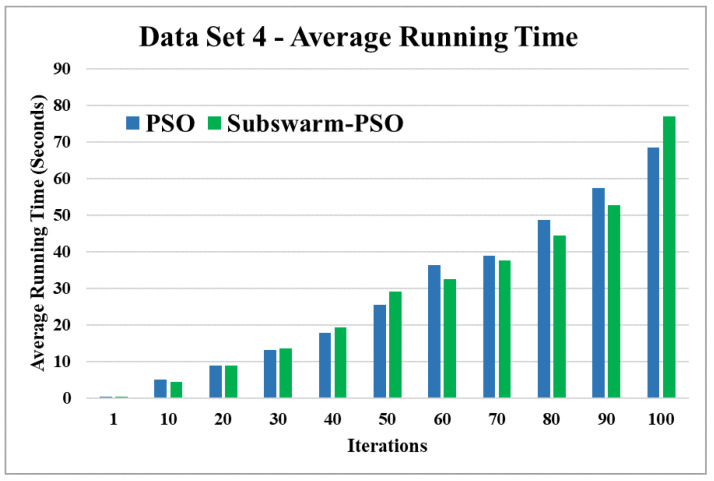
Average running time for data set 4.

**Figure 13 sensors-22-09653-f013:**
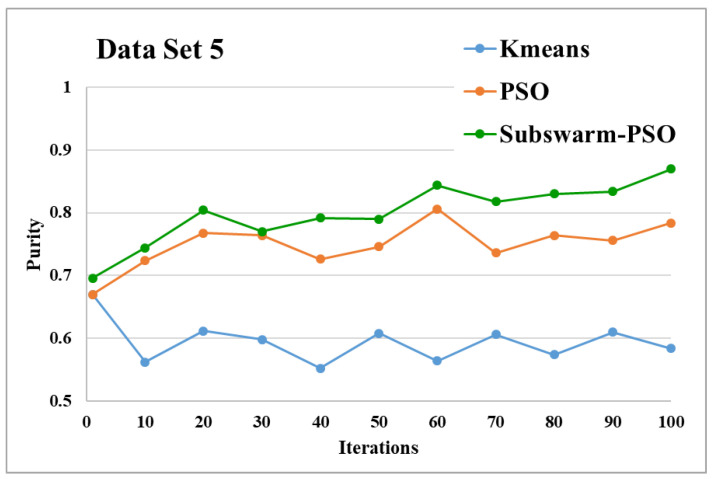
Purity comparison for data set 5.

**Figure 14 sensors-22-09653-f014:**
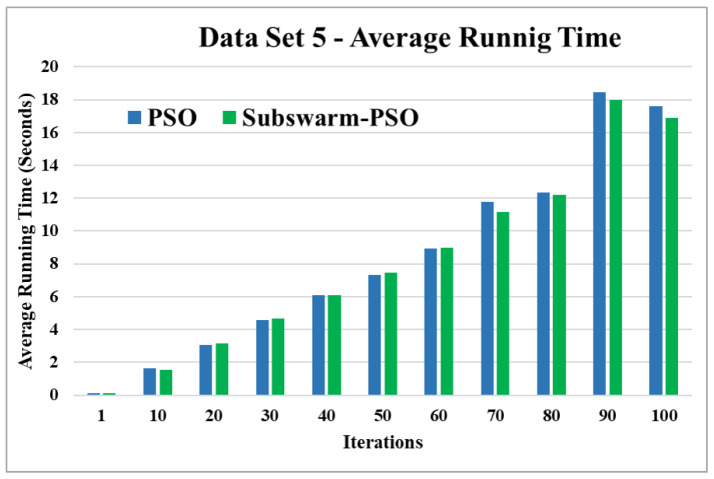
Average running time for data set 5.

**Figure 15 sensors-22-09653-f015:**
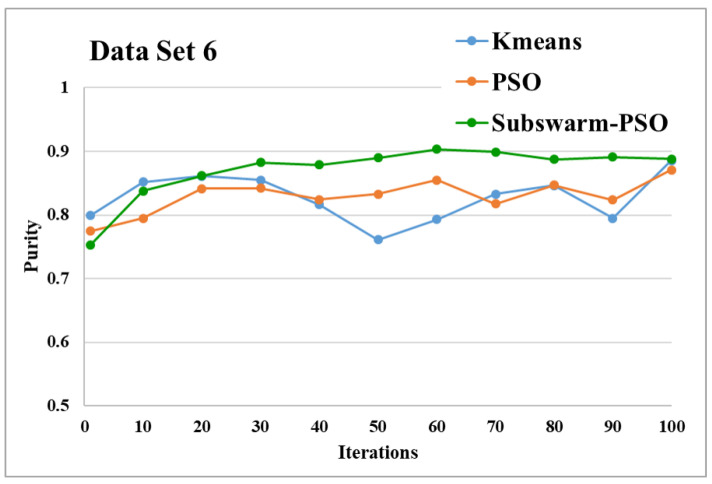
Purity comparison for data set 6.

**Figure 16 sensors-22-09653-f016:**
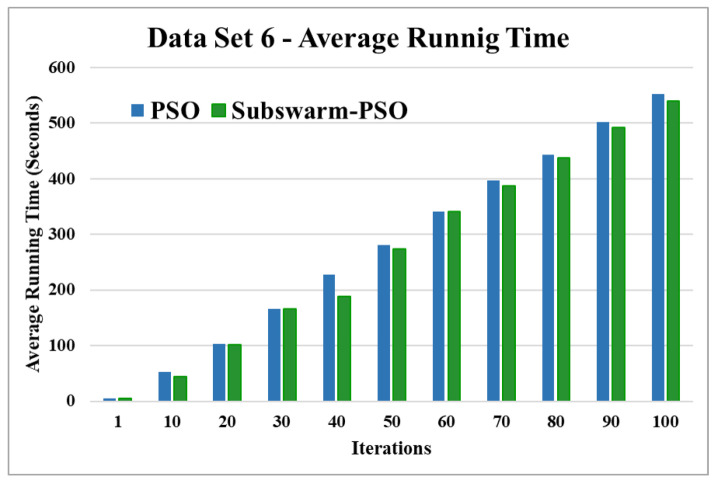
Average running time for data set 6.

**Table 1 sensors-22-09653-t001:** Related research work.

Authors	Approach	Purpose/Results
J. Liang et al. [12]	In the Dynamic multi-swarm PSO (DMS-PSO) algorithm, the swarm population is divided into many small swarms. Based on various regrouping schedules these swarms are frequently regrouped and information among the swarms is exchanged.	Improved performance than the PSO algorithm.
Xu, Y. et al. [13]	Three sub-swarm discrete PSO approach divides the whole swarm into three sub-swarms. Here, one sub-swarm evolves with the standard PSO model, the second sub-swarm with a social-only model, and the third sub-swarm with a cognition-only model.	Able to find the best fitness more quickly and precisely than discrete PSO.
Y. Liu et al. [14]	A modified multi-swarm PSO using sub-swarms and a multi-swarm scheduler.	Used to monitor and control each sub-swarm using the rules to solve the discrete problem.
Ye Wenxing et al. [15]	A novel multi-swarm PSO with a dynamic learning strategy and each sub-swarm is classified into ordinary particles and communication particles. Here, ordinary particles focus on exploitation, and communication particles focus on exploration.	Comparing other algorithms, effectively solving complex problems.
C. Qiu et al. [16]	The novel multi-swarm PSO for feature selection which splits the population of PSO into several small-sized sub-swarms and each sub-swarm updates its positions by the guidance of local best particles from its own sub-swarms.	Finds feature subsets with high classification accuracies and smaller numbers of features.
X. Xia et al. [17]	In the proposed dynamic multi-swarm global PSO, the entire population is divided into a global sub-swarm and multiple dynamic sub-swarms. the global sub-swarm focuses on exploitation and the dynamic multiple sub-swarm focuses on exploration.	Provides more performance in complex problem solving and avoids premature convergence while solving multimodal problems.

**Table 2 sensors-22-09653-t002:** Benchmark data sets.

Data Sets	Source	No. of Documents	No. of Terms	No. of Clusters
1	20 newsgroups	1427	23,057	2
2	BBC Sports	737	13,016	5
3	BBC Sports	40	2596	5
4	20 newsgroups	200	8716	4
5	20 newsgroups	100	5549	3
6	Scientific papers	675	27,416	4

**Table 3 sensors-22-09653-t003:** Different number of abandoned particles with least solutions of subswarm-PSO for 10 particles.

Iterations	Standard PSO			1 Abandoned P			n/3 Abandoned P			n/2 Abandoned P			(n − (n/3)) Abandoned P			n − 1 Abandoned P		
	Max.	Mean	Std.	Max.	Mean	Std.	Max.	Mean	Std.	Max.	Mean	Std.	Max.	Mean	Std.	Max.	Mean	Std.
1	0.690	0.670	0.019	0.740	**0.706**	0.042	0.690	0.660	0.033	0.720	0.668	0.044	0.740	0.664	0.047	0.720	0.682	0.033
10	0.790	0.724	0.043	0.830	0.772	0.046	0.780	0.734	0.029	0.840	**0.806**	0.046	0.830	0.754	0.052	0.820	0.754	0.044
20	0.810	0.768	0.049	0.810	0.780	0.037	0.800	0.770	0.021	0.790	0.776	0.017	0.820	**0.794**	0.024	0.800	0.764	0.030
30	0.840	0.764	0.066	0.860	0.776	0.050	0.880	**0.812**	0.056	0.880	**0.812**	0.041	0.800	0.788	0.008	0.790	0.780	0.010
40	0.840	0.726	0.069	0.780	0.754	0.024	0.820	0.766	0.038	0.860	**0.816**	0.038	0.870	0.802	0.040	0.790	0.776	0.017
50	0.810	0.746	0.049	0.860	0.818	0.036	0.870	0.816	0.034	0.830	0.800	0.021	0.840	**0.820**	0.019	0.800	0.782	0.015
60	0.850	0.806	0.055	0.830	0.772	0.058	0.850	0.796	0.054	0.860	**0.836**	0.025	0.860	**0.836**	0.027	0.830	0.800	0.020
70	0.830	0.736	0.056	0.880	0.834	0.034	0.880	0.828	0.048	0.880	**0.842**	0.031	0.880	0.832	0.038	0.800	0.784	0.021
80	0.870	0.764	0.074	0.850	0.792	0.049	0.850	**0.828**	0.026	0.850	0.822	0.026	0.860	0.818	0.027	0.840	0.800	0.029
90	0.860	0.756	0.063	0.860	0.814	0.050	0.880	0.840	0.032	0.880	**0.848**	0.029	0.850	0.822	0.026	0.830	0.804	0.027
100	0.850	0.784	0.055	0.850	0.798	0.049	0.870	0.816	0.048	0.910	**0.850**	0.052	0.860	0.834	0.027	0.830	0.800	0.019
Average	0.749			0.783			0.788			0.807			0.797			0.775		
Ranks	6			4			3			1			2			5		

**Table 4 sensors-22-09653-t004:** Different number of abandoned particles with least solutions of subswarm-PSO for 20 particles.

Iterations	Standard PSO			1 Abandoned P			n/3 Abandoned P			n/2 Abandoned P			(n − (n/3)) Abandoned P			n − 1 Abandoned P		
	Max.	Mean	Std.	Max.	Mean	Std.	Max.	Mean	Std.	Max.	Mean	Std.	Max.	Mean	Std.	Max.	Mean	Std.
1	0.800	**0.745**	0.037	0.800	0.682	0.068	0.730	0.702	0.036	0.690	0.680	0.012	0.750	0.724	0.026	0.740	0.698	0.032
10	0.925	**0.830**	0.069	0.870	0.796	0.066	0.850	0.800	0.041	0.890	0.820	0.049	0.860	0.794	0.056	0.810	0.780	0.034
20	0.850	0.815	0.029	0.850	0.804	0.051	0.840	0.778	0.055	0.950	**0.862**	0.067	0.880	0.838	0.036	0.850	0.792	0.038
30	0.850	0.795	0.045	0.880	**0.860**	0.016	0.880	0.834	0.050	0.890	0.854	0.038	0.880	0.836	0.033	0.860	0.810	0.043
40	0.925	0.850	0.043	0.860	0.814	0.043	0.900	0.850	0.045	0.880	**0.852**	0.026	0.850	0.828	0.028	0.820	0.804	0.015
50	0.875	0.835	0.034	0.910	**0.860**	0.045	0.860	0.836	0.023	0.900	0.856	0.038	0.860	0.828	0.022	0.820	0.794	0.017
60	0.925	0.810	0.070	0.880	0.824	0.055	0.920	**0.904**	0.011	0.900	0.860	0.028	0.940	0.872	0.054	0.870	0.840	0.024
70	0.900	0.815	0.055	0.880	0.802	0.072	0.880	0.822	0.056	0.920	**0.866**	0.033	0.870	0.822	0.030	0.820	0.802	0.015
80	0.950	0.840	0.076	0.860	0.834	0.022	0.940	**0.886**	0.046	0.920	0.880	0.058	0.920	0.864	0.036	0.850	0.826	0.028
90	0.850	0.825	0.018	0.890	0.874	0.015	0.880	0.850	0.032	0.920	**0.906**	0.022	0.900	0.836	0.036	0.850	0.810	0.029
100	0.875	0.825	0.035	0.870	0.848	0.016	0.920	**0.888**	0.024	0.920	0.840	0.047	0.890	0.864	0.019	0.830	0.824	0.009
Average	0.817			0.818			0.832			0.843			0.828			0.798		
Ranks	5			4			2			1			3			6		

**Table 5 sensors-22-09653-t005:** Maximum, mean, and standard deviation results for all iteration numbers for clustering algorithms for six data sets.

Data Sets	K-Means			PSO			Subswarm-PSO		
	Max.	Mean	Std.	Max.	Mean	Std.	Max.	Mean	Std.
1	0.562	0.527	0.022	0.717	0.686	0.027	0.752	**0.728**	0.020
2	0.887	**0.763**	0.101	0.729	0.680	0.038	0.757	0.721	0.031
3	0.741	0.651	0.081	0.845	0.792	0.046	0.864	**0.820**	0.033
4	0.531	0.450	0.064	0.655	0.612	0.042	0.686	**0.650**	0.028
5	0.655	0.595	0.055	0.822	0.749	0.054	0.845	**0.807**	0.034
6	0.904	0.827	0.070	0.884	0.829	0.046	0.894	**0.888**	0.006

**Table 6 sensors-22-09653-t006:** Ranking table of purity mean values for text document clustering algorithms.

Data Sets	K-Means	PSO	Subswarm-PSO
1	3	2	1
2	1	3	2
3	3	2	1
4	3	2	1
5	3	2	1
6	3	2	1
Total Ranks	3	2	1

## Data Availability

This study used data sets from the publicly archived datasets ucd.ie (accessed on 12 November 2022), uci.edu (accessed on 12 November 2022), and source code from ju.edu.jo (accessed on 12 November 2022).

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
