# Peer review of "Dynamic Sub-Swarm Approach of PSO Algorithms for Text Document Clustering"

_sensors, 2022, doi:10.3390/s22249653_

Round 1
Reviewer 1 Report
The authors may consider changing Tables 2 and 3 to a rotating table format.
Author Response
Dear Editor,
We would like to thank the reviewers and editor for spending their valuable time in reviewing our paper entitled “Dynamic Sub-Swarm Approach of PSO Algorithms for Text
Document Clustering” (Manuscript ID: sensors-2062352). We appreciate the reviewers’ interest in making our manuscript better. The comments from the reviewers and editor are grateful and helped us to improve the quality and readership of our manuscript. Here, we revised our manuscript based on the comments from the reviewers. We hope the revised version is now suitable for publication in the journal ‘Sensors’. We attached our responses to the Reviewer’s comments altogether.
Please note that the reviewer comments are highlighted with ‘bold text in dark red color’ and our responses are given in ‘normal text in black fonts’. The modifications made in the text are highlighted as ‘italicized black fonts’.
Reviewer 1
Comments and Suggestions for Authors:
The authors may consider changing Tables 2 and 3 to a rotating table format.
Response:
We thank the reviewer for the valuable comment. The tables 2 and 3 are rotated to 90 degrees as suggested [References: Page: #10 and #11].
Reviewer 2 Report
The manuscript has many symbols, thereby, may benefit from a nomenclature list that defines all symbols at the beginning of the manuscript. Please create a nomenclature that contains all the symbology used in the manuscript.
Ln.22 Please change the sentences as; the most popular algorithms for clustering are K-means and its variants, as the K-means algorithm is a simple and most used unsupervised partitioning algorithm [4].
Ln.36 Please elaborate on your dynamic subswarm-PSO algorithm. Briefly answer the question; Why do we need this?
Ln.39 Please use not subjective but objective words. Share brief statistical data for the average execution time.
Ln.47 Please supply a table (in Section 2) that shows the differences between previous studies column by column. The table would be useful for organising and analysing your literature review. Otherwise, the section ‘Related Work’ will appear as a bunch of words.
Ln. 69 Section 3 attempts to explain the mechanism behind Figure 3. But there is no ‘Figure 1’ as a word in all text. Give a quick explanation of Figure 1.
Ln 86 Clearly explain all steps in Figure 1. For example, Where is validation? Check all steps and explain them step by step (what are they? why are used? any current example etc.)
Ln 99, the given algorithm needs a caption.
Ln 105, the PSO algorithm needs a caption.
Ln 149, the proposed algorithm needs a caption.
Ln 149, İn the ‘for loop’ of Algorithm 3, if we use ‘j’ for SP1, please add it to ‘after randomInitialization()’.
It appears that six different benchmark machine learning data sets are used to discuss the performance of the study. If any limitation exists, please indicate.
Please create a conclusion chapter (not a summary) that includes a future word in this field. The potation application areas also should be mentioned.
Author Response
Dear Editor,
We would like to thank the reviewers and editor for spending their valuable time in reviewing our paper entitled “Dynamic Sub-Swarm Approach of PSO Algorithms for Text
Document Clustering” (Manuscript ID: sensors-2062352). We appreciate the reviewers’ continuous interest in making our manuscript better. The comments from the reviewers and editor are grateful and helped us to improve the quality and readership of our manuscript. Here, we revised our manuscript based on the comments from the reviewers. We hope the revised version is now suitable for publication in the journal ‘Sensors’. We attached our responses to the Reviewer’s comments altogether.
Please note that the reviewer comments are highlighted with ‘bold text in dark red color’ and our responses are given in ‘normal text in black fonts’. The modifications made in the text are highlighted as ‘italicized black fonts’.
Reviewer 2
Comments and Suggestions for Authors:
The manuscript has many symbols, thereby, may benefit from a nomenclature list that defines all symbols at the beginning of the manuscript. Please create a nomenclature that contains all the symbology used in the manuscript.
Response: We thank the reviewer for the valuable comment. As suggested by the reviewer, the nomenclature list is added in this paper [Page: #20~#21, Lines: #360 ~ #382].
Ln.22 Please change the sentences as; the most popular algorithms for clustering are K-means and its variants, as the K-means algorithm is a simple and most used unsupervised partitioning algorithm [4].
Response: We thank the reviewer for this gentle mention. The sentences are changed as “The most popular algorithms for clustering are K-means and its variants, as the K-means algorithm is a simple and most used unsupervised partitioning algorithm” as suggested by the reviewer [Page: #1, Lines: #22~#24].
Ln.36 Please elaborate on your dynamic subswarm-PSO algorithm. Briefly answer the question; Why do we need this?
Response: We thank the reviewer for this constructive comment which increases the quality of manuscript.
As suggested by the reviewer, the below lines were added.
“The PSO provides an effective solution to text document clustering [6]. However, the PSO algorithm usually suffers from falling into a premature convergence to local optimum [7]. This is because PSO initializes the particles in starting of the algorithm and the searching behavior includes that all the particles move towards the best solution and searches around the local area. In this case, if the initialization does not explore the proper area including global solution, there is no option to research the undiscovered area to globally search again in the middle of the algorithm. To improve the global searching capability of PSO algorithm for text document clustering, in this paper, a dynamic subswarm-PSO algorithm is proposed [8]. This proposed algorithm will reinitialize the number of worst fitness particles in each iteration.”
[Pages: #1~#2, Lines: #30~#40]
Ln.39 Please use not subjective but objective words. Share brief statistical data for the average execution time.
Response: We thank the reviewer for the constructive comment. The statistical data are added for the average execution time. [Pages: #12~#13; Lines: #277, #283, #289, #294, #298, and #302]
Ln.47 Please supply a table (in Section 2) that shows the differences between previous studies column by column. The table would be useful for organising and analysing your literature review. Otherwise, the section ‘Related Work’ will appear as a bunch of words.
Response: We thank the reviewer for the valuable comment. As suggested by the reviewer, the table 1 is added for the “Related work” section to clearly show the differences between previous studies. [Page: #2, Section: #2]
Ln. 69 Section 3 attempts to explain the mechanism behind Figure 3. But there is no ‘Figure 1’ as a word in all text. Give a quick explanation of Figure 1.
Ln 86 Clearly explain all steps in Figure 1. For example, Where is validation? Check all steps and explain them step by step (what are they? why are used? any current example etc.)
Response: We thank the reviewer for the valuable comments. Following the reviewer’s comment, the section 3 has been modified to elaborate the process of text document clustering as follows,
“This section describes the main process of text document clustering. This process includes text document collection, pre-processing, document representation, clustering, and cluster validation as shown in the Figure 1 [18].
The pre-processing is an important step to enhance the performance of the clustering algorithm. As shown in the Figure 1, after collecting the required raw text documents, the text pre-processing is used to clean these text documents by applying natural language processing techniques such as tokenization, stop word removal, stemming, and term weighting to delete the unwanted data and manipulate the data. Then, pre-processing turns this clean documents into t × d term-document matrix. Here, t represents the number of unique terms in the document collection, and d represents the number of documents. The text document clustering algorithms (PSO and subswarm PSO) directly uses this matrix to convert the document data sets into meaningful sub-collections. As shown in the Figure 1, to validate the quality of results from text document clustering algorithms, a few cluster evaluation metrics can be used. The next subsection explains the evaluation metrics in detail.”
[Pages: #2~#3, Lines: #56~#70]
Ln 99, the given algorithm needs a caption.
Ln 105, the PSO algorithm needs a caption.
Ln 149, the proposed algorithm needs a caption.
Response: We thank the reviewer for the valuable comments. As suggested by the reviewer the captions are changed as follows.
[Page: #4, Line: #90] From “4.1. K-means” to “4.1. K-means Algorithm for Text Document Clustering”
[Page: #4, Line: #96] From “4.2. PSO Algorithm” to “4.2. PSO Algorithm for Text Document Clustering”
[Page: #6, Line: #140] From “6. Algorithm of Proposed Approach” to “6. Proposed Dynamic Sub-swarm PSO Algorithm for Text Document clustering”
Ln 149, İn the ‘for loop’ of Algorithm 3, if we use ‘j’ for SP1, please add it to ‘after randomInitialization()’.
Response: We thank the reviewer for the valuable comment. Following the reviewer’s comment the line changed into “SP1[j]← randomInitialization();”in the algorithm [Page:#6, Line: #140]
It appears that six different benchmark machine learning data sets are used to discuss the performance of the study. If any limitation exists, please indicate.
Response: Although there are no limitations are expected in terms of number of data sets or the total size of each data set, considering the computation time and our resources, we have used only six different data sets for this study. This has been highlighted in the manuscript as “Considering the computation time and our resources, we have used only six different data sets for this study.” [Pages: #7, Lines: #170~#171]
Please create a conclusion chapter (not a summary) that includes a future word in this field. The potation application areas also should be mentioned.
Response: We thank the reviewer for the valuable comment. As suggested by the reviewer, the name of the section 9 is changed from “Summary” to “Conclusion” and add the below lines in to the conclusion section to address the domain.
“Previous literature shows that the PSO is also performing well in a few other problems such as clustering, feature selection, and scheduling [18]. Our proposed approach also may perform well in those problems as PSO algorithm. As a future study, we are aiming to apply and check the performance of the proposed subswarm-PSO approach in different domains.”
[Page: #20; Lines: #342~#346]
Reviewer 3 Report
The authors have proposed a better approach to improve the efficiency of the standard Particle Swarm Optimization technique for text document clustering applications. The changes required from the author's end have been highlighted in the manuscript.
1. No studies show that the standard PSO converges to local optima.
2. In the present study, there is no solid evidence to show that PSO failed to capture the global optima.
3. Hyperparameters play a very important role in the convergence of the solution and computing time. There are no details about the hyperparameters of standard PSO and improved PSO used for the present study.
4. From the results, it is observed that there is not much difference in computation time between standard PSO and modified PSO. The authors claim that the modified PSO is better than the standard PSO.
5. No sensitivity study shows the efficiency of modified PSO compared to standard PSO.

Author Response
Dear Editor,
We would like to thank the reviewers and editor for spending their valuable time in reviewing our paper entitled “Dynamic Sub-Swarm Approach of PSO Algorithms for Text
Document Clustering” (Manuscript ID: sensors-2062352). We appreciate the reviewers’ continuous interest in making our manuscript better. The comments from the reviewers and editor are grateful and helped us to improve the quality and readership of our manuscript. Here, we revised our manuscript based on the comments from the reviewers. We hope the revised version is now suitable for publication in the journal ‘Sensors’. We attached our responses to the Reviewer’s comments altogether.
Please note that the reviewer comments are highlighted with ‘bold text in dark red color’ and our responses are given in ‘normal text in black fonts’. The modifications made in the text are highlighted as ‘italicized black fonts’.
Reviewer 3
Comments and Suggestions for Authors:
- No studies show that the standard PSO converges to local optima.
Response: We thank the reviewer for the valuable comment. Following the reviewer comment, the following reference is added which shows PSO converges to local optima.
“However, the PSO algorithm usually suffers from falling into a premature convergence to local optimum [7].”
[Page: #1; Line: #31~#32]
“Salehizadeh, S.; Yadmellat, P.; Menhaj, M. Local Optima Avoidable Particle Swarm Optimization. In Proceedings of the 2009 IEEE Swarm Intelligence Symposium, 2009, pp. 16–21. https: //doi.org/10.1109/SIS.2009.4937839.”
- In the present study, there is no solid evidence to show that PSO failed to capture the global optima.
Response: We thank the reviewer for the valuable comment. Following the reviewers comments, we have added the below literature to the manuscript in the below line and this literature shows that there is no guarantee that PSO capture the global optima all the time.
“To improve the global searching capability of PSO algorithm for text document clustering, in this paper, a dynamic subswarm-PSO algorithm is proposed [8].”
[Page:#2; Lines: #37~#39]
“Mei, C.; Liu, G.; Xiao, X. Improved particle swarm optimization algorithm and its global convergence analysis. In Proceedings of the 2010 Chinese Control and Decision Conference, 2010, pp. 1662–1667. https://doi.org/10.1109/CCDC.2010.5498348.”
- Hyperparameters play a very important role in the convergence of the solution and computing time. There are no details about the hyperparameters of standard PSO and improved PSO used for the present study.
Response: We thank the reviewer for the constructive comment, which increases the quality of manuscript. As suggested by the reviewer the following lines were added in the manuscript
“Here, to compare the performance of PSO and proposed subswarm-PSO, the same default parameters ω= 0.9, c1 = 0.5, and c2 = 0.3 of PSO algorithm were used in both algorithms from the literature 17,24].
[Page: #8; Lines: #197~#199]
- From the results, it is observed that there is not much difference in computation time between standard PSO and modified PSO. The authors claim that the modified PSO is better than the standard PSO.
Response: The experimental results show that our proposed dynamic subswarm-PSO takes “5.1%, 0.8%, 13.7%, 2.1%, 0.75%, and 4.31%” lesser execution time than the standard PSO algorithm.
However, the differences are not much as mentioned in the reviewer’s comment. To address this comment, the following changes have been done in the manuscript.
- The text mentioned in manuscript is changed from “execution time of our proposed algorithm is less than the standard PSO algorithm” to “the average execution time of our proposed algorithm is little less than the standard PSO algorithm”. [Page: #2; Line: #44]
- The statistical data are added for the average execution time in [Pages: #12~#13; Lines: #277, #283, #289, #294, #298, and #302] instead of mentioning subjective words.
5. No sensitivity study shows the efficiency of modified PSO compared to standard PSO.
Response: In this study, we attempt to improve the performance of PSO algorithm by changing the flow of an algorithm. Here, we use same parameters in both algorithms to compare the performance instead of concentrating on parameter sensitivity study. We hope that a comparative sensitivity study is of our future study.
- Highlighted comments in manuscript
Response: We thank the reviewer for the valuable comments that are highlighted in the manuscript. We corrected all the highlighted changes appropriately in the updated version.
Round 2
Reviewer 3 Report
All the reviews are addressed by the authors. It can be accepted for publication.